# Advanced Glycation End-Products and Diabetic Neuropathy of the Retina

**DOI:** 10.3390/ijms24032927

**Published:** 2023-02-02

**Authors:** Toshiyuki Oshitari

**Affiliations:** 1Department of Ophthalmology and Visual Science, Chiba University Graduate School of Medicine, Inohana 1-8-1, Chuo-ku, Chiba 260-8670, Japan; tarii@aol.com; Tel.: +81-43-226-2124; Fax: +81-43-224-4162; 2Department of Ophthalmology, School of Medicine, International University of Health and Welfare, 4-3 Kozunomori, Narita 286-8686, Japan

**Keywords:** diabetic retinopathy, neurovascular unit, advanced glycation end-products, neuroprotection, receptor of AGEs, polyol pathway, oxidative stress, apoptosis, anti-glycation, nutrients

## Abstract

Diabetic retinopathy is a tissue-specific neurovascular impairment of the retina in patients with both type 1 and type 2 diabetes. Several pathological factors are involved in the progressive impairment of the interdependence between cells that consist of the neurovascular units (NVUs). The advanced glycation end-products (AGEs) are one of the major pathological factors that cause the impairments of neurovascular coupling in diabetic retinopathy. Although the exact mechanisms for the toxicities of the AGEs in diabetic retinopathy have not been definitively determined, the AGE-receptor of the AGE (RAGE) axis, production of reactive oxygen species, inflammatory reactions, and the activation of the cell death pathways are associated with the impairment of the NVUs in diabetic retinopathy. More specifically, neuronal cell death is an irreversible change that is directly associated with vision reduction in diabetic patients. Thus, neuroprotective therapies must be established for diabetic retinopathy. The AGEs are one of the therapeutic targets to examine to ameliorate the pathological changes in the NVUs in diabetic retinopathy. This review focuses on the basic and pathological findings of AGE-induced neurovascular abnormalities and the potential therapeutic approaches, including the use of anti-glycated drugs to protect the AGE-induced impairments of the NVUs in diabetic retinopathy.

## 1. Introduction

Diabetes is a metabolic disease characterized by chronic hyperglycemia, and the prevalence of diabetic patients has increased to approximately 600 million and will reach 700 million worldwide by 2045 [1]. Diabetic retinopathy is a major complication in diabetic patients, and it is defined as a tissue-specific neurovascular impairment of the interdependence between multiple cells, which consists of the neurovascular units (NVUs) in both type 1 and type 2 diabetes [2]. Twenty years after the onset of diabetes, all patients with type 1 and 60% of patients with type 2 diabetes will have progressed to diabetic retinopathy [3]. As a result, 95 million diabetic patients have diabetic retinopathy in the world [4], and 10% of the patients with diabetes have progressed to the vision-threatening stages of diabetic retinopathy. This includes clinically significant diabetic macular edema and proliferative diabetic retinopathy [5].

The NVUs in the retina is composed of the vascular cells (endothelial cells and pericytes), the glial cells (Müller cells, astrocytes, and microglia), and the neuronal cells (ganglion cells, amacrine cells, bipolar cells, and horizontal cells) [6,7,8,9]. The photoreceptor cells are mainly nourished by nutrients that pass through the choroidal membrane and retinal pigment epithelium. Thus, the interdependence between photoreceptors and the NVUs may be weaker than other neurons in the retina. The interdependence of the cells constituting the NVUs maintains retinal homeostasis and function under healthy conditions.

Chronic hyperglycemia is a trigger for developing an impairment of the interdependence of cells which consists of the NVUs, because chronic hyperglycemia causes an increase in several stress factors, including advanced glycation end-products (AGEs), inflammation, oxidative stress, activation of the polyol pathway and protein kinase C (PKC). The increase in these stress factors leads to microvascular impairments and glial dysfunction, which then results in increasing the expression of vascular endothelial growth factor (VEGF) and inflammatory cytokines in the eye. This is followed by a breakdown of the blood–retinal barrier [6,7,8,9]. During the course of these changes, there is also excitotoxicity and a decrease in neurotrophic factors resulting in neuronal cell death and axon degeneration, which leads directly to a reduction in vision [6,7,8,9].

Among these stress factors, oxidative stress causes differences in post-transcriptional RNA editing events and gene expression, which may be related to pathological changes in diabetic retinopathy, including retinal degeneration and neovascularization [10,11,12]. The main targets of oxidative stress are glucose, amino acids, and lipids, which result in generating reactive carbonyl compounds (RCOs) [13]. These increasing RCOs, which cause carbonyl stresses, facilitate the synthesis of AGEs under diabetic conditions [13].

The AGEs are pathological factors involved in the onset and progression of diabetic retinopathy and may be one of the therapeutic targets to block to ameliorate the impairment of the NVUs in diabetic retinopathy [14]. This review focuses on the basic and pathological findings of AGEs and potential therapeutic approaches, including the identification of anti-glycated drugs to protect the AGE-induced neurovascular impairments.

## 2. AGE Production and Accumulation

The process for the formation of AGE in the body is complicated and still not completely determined, but the early stage of AGE formation is known to be the Maillard reaction. The Maillard reaction is a nonenzymic reaction (Figure 1).

The intermediate product of the process of AGE formation is an Amadori compound. Amadori compounds are relatively stable and can be precursors of AGEs [15]. The process from a Shiff base to an Amadori compound is thought to be irreversible. Glycohemoglobin A1c (HbA1c) is one of Amadori compounds, which is a clinical marker of glucose control before 1–2 months in diabetic patients.

Amadori products are sources of the α-dicarbonyl compounds via metal ions catalyzed autoxidation of glucose or the nonoxidative hydrolysis and fragmentation. 3-deoxy-D-erythro-hexos-2-ulose (3-deoxyglucosone; 3-DG) is a 10,000 times stronger reactive compound for AGE formation than glucose. 3-DG is known to be synthesized from fructose or fructose-3-phosphate via the polyol pathway [16].

The final step of AGE formation is very complex, but a possible process is displayed in Figure 2. Other AGEs that have been identified in the body are carboxymethyl arginine (CMA) [17], imidazolone [18], formyl threosyl pyrrole (FTP) [19], argpyrimidine [20], pentosidine [21], and crossline [22].

The sources of AGEs, including chronic hyperglycemia, are not only endogenous but also exogenous, including foods and tobacco. For example, the smoke of tobacco contains glyoxal and methylglyoxal, and their presence results in the accumulation of AGEs in the body [26]. After intake of AGEs from foods, 10–30% of AGEs are absorbed, and one-third of the AGEs intake is excreted. Thus, 6–7% of AGEs in foods are expected to accumulate in the body. A recent review summarizes the high- and low-AGE food products. High AGE levels of foods are foods rich in protein and fat, baked and grilled food, frying products, and animal products. On the other hand, foods low in AGE are low-fat products, high-carbohydrates products, raw products, and products cooked at low temperatures [26]. Therefore, eating low-level AGE foods may be one method of preventing the progression of diabetic conditions.

The polyol pathway is one of the main sources of AGE formation. The polyol pathway is activated under chronic hyperglycemic conditions (Figure 3). Briefly, under high glucose conditions, excess glucose taken in by retinal cells is metabolized into sorbitol by aldose reductase, and sorbitol is metabolized into fructose by sorbitol dehydrogenase [27,28]. The fructose is converted into fructose-3-phosphate and 3-DG. Because 3-DG is a 10,000 times stronger reactive compound for AGE production than glucose, AGE formation is accelerated by increasing 3-DG by activating the polyol pathway (Figure 3) [28]. During the activation of the polyol pathway, the level of an antioxidant molecule, nicotinamide adenine dinucleotide phosphate (NADPH), is decreased because of its excessive consumption. In addition, nicotinamide adenine dinucleotide (NADH) is increased, which results in PKC activation. These changes contribute to the increased production of reactive oxygen species (ROS) [27,28]. The excess ROS production causes inflammation and cell death in diabetic retinopathy [29]. At the same time, sorbitol is a hyperosmotic factor that increases osmotic stress. The increase in sorbitol production may be due to Müller cell swelling in early diabetic retinopathy (Figure 3) [30]. Oxidative stress induces the alterations of several transcription factors, such as the activator protein-1, p53, and nuclear factor-kappa B (NF-κB) [31]. However, the alteration of gene expression induced by oxidative stress is beyond the scope of this review.

## 3. Pathophysiology of AGEs

An increase in the synthesis of AGEs leads to biochemical damage to the retinal tissue. The biochemical damages are induced by both the receptors of AGE (RAGE)-dependent and RAGE-independent mechanisms [9]. Under physiological conditions, RAGE is expressed in many tissues and cells, including neurons, glial cells, smooth muscle cells, fibroblasts, monocytes, and endothelial cells [32,33,34]. Not only AGEs but also other ligands such as S100, amphoterin, macrophage-1 antigen, and amyloid-β can bind to the RAGEs, which results in pathophysiological events, including inflammation, ROS generation, and cell death [35].

Although not all cellular mechanisms are activated under the AGE-RAGE axis [36], the AGE/RAGE interaction is thought to activate several cellular pathways and to be associated with the pathophysiology of diabetic retinopathy [9]. For example, the NADPH- NF-κB axis, the RAS-extracellular signal-regulated kinase 1/2 (ERK1/2) axis, and the Ras-mitogen-activated protein kinase (MAPK) axis are involved in inflammatory reactions [36,37,38]. The Ras-c-Jun N terminal kinase (JNK) axis is related to apoptosis.

An increase in the synthesis of the AGEs leads to cross-link formations with proteins, which result in reducing the protein turnover and energy production, activation of endoplasmic reticulum (ER) stress, and macrophage activation [9,39,40]. In addition, the Amadori compounds can become the source of ROS. These cellular events are involved in the AGE-RAGE independent mechanisms (Figure 4).

Ying et al. used the non-invasive skin autofluorescence method [41] to examine the association between the AGE level in the skin and the stages of diabetic retinopathy [42]. They concluded that the AGE level in the skin was associated with the prevalence and severity of diabetic retinopathy and that the AGE level in the skin may be a more suitable indicator than HbA1c for chronic hyperglycemic controls [42]. Similarly, Zhang et al. suggest that the AGE level in the skin is significantly associated with the DR stage in type 2 diabetes mellitus [43]. Takayanagi et al. examined the level of skin autofluorescence to assess the AGE level in the body, and they reported that it was higher in diabetic patients than in non-diabetic patients. They concluded that the higher AGE scores were independently associated with the progression of diabetic retinopathy [44]. In addition, higher AGE scores were risk factors for proliferative diabetic retinopathy [44]. They also concluded that the AGE scores in the skin could be obtained non-invasively, and it was a reliable marker of patients developing sight-threatening diabetic retinopathy [44].

AGE accumulation was found in the endothelial cells of the retina [45], which resulted in capillaries occlusion through increased levels of intracellular cell adhesion molecules followed by culminating retinal ischemic injury [46]. Furthermore, the AGEs in pericytes facilitate pericyte apoptosis by the activation of caspase-3 due to a reduction in the Bcl/Bax ratio [47]. Pericyte loss can trigger microvascular abnormalities, including endothelial cell proliferation and dysfunction in diabetic retinopathy [9,48]. A recent study showed that the higher AGEs score in the skin was a risk factor for diabetic retinopathy, but the score of the carotenoids, major antioxidants, was significantly and negatively correlated with the AGE score, which reflected the vegetable uptake score [49]. Oxidative stress is associated with the pathogenesis of AGEs in diabetic retinas (Figure 4), and carotenoids are plant-derived antioxidants. Shalini et al. suggested that decreased plasma levels and lower dietary intake of carotenoids were found in patients with diabetic retinopathy compared to those without diabetic retinopathy [50].

## 4. AGEs and Retinal Neuronal Abnormalities

One of the best-known AGEs in the body is CML [23]. However, the results of earlier studies indicated that CML is not strongly associated with the severity of diabetic complications [51,52]. Thus, non-CML AGEs are more important for the pathogenesis of diabetic complications. On the other hand, glycolaldehyde- and glyceraldehyde-derived AGEs induce pericyte loss and increased expression of VEGF, and they promote endothelial cell proliferation. These changes result in the progression of diabetic retinopathy [53,54]. Furthermore, glyceraldehyde-derived AGEs are more toxic than glucose-derived AGEs, and they induce neuronal cell death in cortical neurons [55,56]. However, no studies have demonstrated the toxic effects of glycolaldehyde- and glyceraldehyde-derived AGEs on retinal neurons. Thus, we have investigated the toxic effects of aldehyde-derived AGEs on neuronal cell death and regeneration of retinal ganglion cells in retinal cultures [57,58]. Three-dimensional collagen gel culture systems were helpful for examining the neurotoxic, neuroprotective, and regenerative effects of factors or reagents of interest [59,60,61,62,63,64]. First, we incubated 10 μg/mL of the standard glucose-derived AGEs, glycolaldehyde-derived AGEs, and glyceraldehyde-derived AGEs in serum-free media. In diabetic patients, between 1 and 120 μg/mL of AGEs (approximately 4–480 μg/mL of glycated bovine serum albumin) are circulating in the body [65]. Previously, 100 μg/mL and 250 μg/mL glucose-derived AGEs were shown to have neurotoxic effects [66]. The results of our study indicated that even a low concentration of AGEs induced neuronal apoptosis and decreased neurite regeneration in cultured retinas [57]. Although there were no differences in the toxic effects among the three types of AGEs, the neuroprotective and regenerative effects of neurotrophin-4 (NT-4) in glycolaldehyde- and glyceraldehyde-derived AGEs incubated retinas were less than those in glucose-derived AGEs incubated retinas [57]. The results of our other study suggested that glycolaldehyde- and glyceraldehyde-derived AGEs reduced the neuroprotective and regenerative effects of neurotrophic factors, NT-4, glial cell line-derived neurotrophic factor, hepatocyte growth factor, and tauroursodeoxycholic acid (TUDCA) more than the glucose-derived AGEs [58]. Taken together, aldehyde-derived AGEs may reduce the therapeutic effects of neurotrophic factors in diabetic retinas.

The AGEs–RAGE axis is associated with the pathogenesis of diabetic complications, including diabetic retinopathy [67,68]. However, in cultured retinas, the neuroprotective and regenerative effects of the RAGE inhibitors were significantly weaker than the other neurotrophic factors, NT-4, TUDCA, and citicoline [64]. Thus, not only the RAGE-dependent pathways but also the RAGE-independent pathways should be involved in the pathogenesis of neuronal abnormalities in diabetic retinas (Figure 4). On the other hand, the AGE/RAGE signaling upregulates NF-κB and VEGF, which then results in vascular abnormalities in diabetic retinas [68]. The results of our previous study indicated that the expression of NF-κB was increased in AGEs incubated retinas, and its expression was correlated with an increase in neuronal cell death in cultured retinas [58] (Figure 4). Thus, inhibition of the AGE/RAGE signaling may be one of the therapeutic options to prevent the progression of diabetic retinopathy.

## 5. Potential Anti-AGEs Therapies for Diabetic Retinopathy

Anti-glycation or anti-AGEs therapies are becoming areas of great interest for the treatment of diabetic complications, including diabetic retinopathy. The levels of exogenous AGEs can be reduced by saving AGEs-rich foods and stopping smoking. However, chronic hyperglycemia leads to the accumulation of endogenous AGEs as described, and thus, anti-glycation or anti-AGEs therapies are needed to reduce the endogenous AGEs in the body. The ideal anti-AGE drugs should have multiple effects, including inhibition of AGEs formation, breakers of cross-links, antioxidants, RAGE antagonists, metal chelating, and carbonyl compound scavengers. Although no ideal anti-AGEs drug has been developed, there are several anti-AGEs drugs, such as chemical drugs, natural drugs, or clinical drugs and natural compounds that have already been used.

### 5.1. AGE Inhibitors

The ideal AGE inhibitor should contain all of the effective factors, e.g., antioxidants, metal chelators, and carbonyl scavengers. Because the Fenton reaction is involved in the late stage of AGE formation, metal chelating reagents are candidates for AGE inhibitors [69]. Although there are no ideal AGE inhibitors, some candidates, both chemical and natural AGE inhibitors, have been suggested.

Aminoguanidine is one of the old and well-known chemical AGE inhibitors. Aminoguanidine acts as a dicarbonyl scavenger, a free radical scavenger, and a metal chelator [70,71]. Two clinical trials have been performed that examined the effects of aminoguanidine on diabetic nephropathy in patients with type 1 (ACTION I) [72] and type 2 (ACTION II) [73]. The ACTION II study was suspended because of severe side effects, including infection, liver dysfunction, lupus-like symptoms, and ANCA-associated vasculitis. The ACTION I group completed the study, but similar side effects were observed. However, the primary end point, which was the time to double the creatinine level in the serum, was not reached [72]. On the other hand, one of the secondary end points, the evaluation of retinopathy progression, was achieved, i.e., aminoguanidine delayed the progression determined by the Early Treatment of Diabetic Retinopathy Study score [72]. However, the effects of aminoguanidine on delaying the progression of diabetic retinopathy were not repeated. An AGE inhibitor, OPB-9195, was developed in Japan to reduce the progression of nephropathy by inhibiting the synthesis of tumor growth factor-β. However, the trial was stopped for unknown reasons [74]. LR-90 is a new AGE inhibitor whose inhibitory effects are stronger than aminoguanidine and pyridoxamine in vitro [75]. LR-90 is thought to be a dicarbonyl scavenger as well as a metal chelator. A recent study showed that LR-90 reduced the acellular capillary numbers and pericyte loss in diabetic animal models, and the authors concluded that LR-90 could be a drug for preventing retinopathy in diabetic patients [76].

Pyridoxamine, an amine of vitamin B6, is a well-known natural AGE inhibitor. It acts as a dicarbonyl scavenger [77], an antioxidant [77,78], and a metal chelator [79,80]. Stitt et al. demonstrated that pyridoxamine reduced capillary drop-out and the expression of extracellular matrix genes in the retinal vessels of diabetic rats. They concluded that pyridoxamine might be useful for treating diabetic retinopathy [81]. Although the primary end points could not be reached, pyridoxamine was shown to be safe and tolerable in two clinical trials in patients with diabetic nephropathy [82,83]. Pramanik et al. examined the effects of vitamin B, C, and E supplementation on the progression of diabetic retinopathy [84]. The clinical trial included 175 patients with type 2 diabetes, and the follow-up period was five years. The results indicated that the supplementation delayed the development of diabetic retinopathy and reduced pathological biomarkers such as ROS, malondialdehyde, AGEs, and VEGF [84]. Recent studies have focused on the neuroprotective effects of pyridoxamine on diabetic retinopathy. Ren et al. showed that pyridoxamine had a protective effect on the photoreceptors of diabetic mice [85]. This protection was accomplished by an ERK /nuclear factor erythroid 2-related factor 2/apoptosis signal-regulating kinase 1 signaling pathway [85]. The results of another study showed that the glyceraldehyde-derived AGEs reduced the neurite outgrowth in human neuroblastoma SH-SY5Y cells, but aminoguanidine and pyridoxamine ameliorated the suppression of neurite outgrowth induced by glyceraldehyde-derived AGEs [86]. Thus, aminoguanidine and pyridoxamine may prevent the development of neuronal abnormalities in diabetic retinopathy. Further studies are needed to determine more conclusively the neuroprotective and regenerative effects of aminoguanidine and pyridoxamine in diabetic retinas. Pyridoxamine is a natural compound, and Japanese sushi toppings include abundant amounts of pyridoxamine [87].

Benfotiamine is a lipophilic derivative of thiamine (vitamin B1) which is absorbed in the body more easily than thiamine. Benfotiamine inhibits the hexosamine pathway, AGE formation, and the DAG-PKC pathway by activating transketolase [88]. In addition, the number of acellular vessels was reduced by inhibiting these three pathways and also by the activation of NF-κB in the retinas of diabetic animals [88]. Clinical trials for examining the effects of benfotiamine on diabetic neuropathy showed positive results in which the neuropathy scores were significantly improved without any severe side effects. Unfortunately, the followed-up periods were short [89,90]. A long-term clinical trial that assesses the length of the corneal nerve fibers by corneal confocal microscopy is ongoing (DRKS00014832) [91]. Clinical trials on the effects of benfotiamine on diabetic retinopathy have not been performed. However, the results of recent studies showed that patients with diabetic polyneuropathy had significantly thinner retinal nerve fiber layers than patients without diabetic polyneuropathy. In addition, the parameters of the ganglion cell complex in early diabetic retinopathy were distinctly abnormal in patients with diabetic polyneuropathy [92,93,94,95]. These results suggested that the pathogenesis of diabetic polyneuropathy is, in part, similar to that of diabetic neuropathy in the retina. Thus, drugs effective for diabetic polyneuropathy may be effective in preventing or slowing the development of diabetic neuropathy in the retina. Further studies are needed to examine the effect of benfotiamine on early diabetic retinopathy.

*Trapa bispinosa* roxb., the water chestnut, is a well-known small herb that contains a high quantity of minerals, ions, and vitamins [96]. Previous in vitro studies showed that *trapa bispinosa* roxb. had significant antioxidative effects against free radicals [97], and it could inhibit the synthesis of CML and the cross-linking of AGEs [98]. It can also degrade the α-dicarbonyl compounds [98]. Recent in vivo studies showed that the systemic administration of *trapa bispinosa* roxb. and lutein reduced the accumulation of AGEs in the retina of streptozotocin-induced diabetic rats [99]. It also improved the abnormal regulation of retinal blood flow in type 2 diabetic mice [100]. In addition, *trapa bispinosa* roxb. and lutein reduced the expression of the glial fibrillary acid protein in the endfeet of Müller cells and also reduced the expression of VEGF in the retinas of diabetic mice [100]. The authors concluded that *trapa bispinosa* roxb. and lutein improved the abnormal neurovascular coupling in the retinas of diabetic mice [100]. Extracts of *trapa bispinosa* roxb. mixed with lutein and zeaxanthin have become commercially available in Japan, but no clinical trials have been performed to evaluate their effects on preventing or slowing the progression of diabetic retinopathy. Thus, clinical studies are needed to determine the effectiveness of these supplements for the early treatment of patients with diabetic retinopathy.

### 5.2. AGE Cross-Link Breakers

Once the AGEs are synthesized, the AGE inhibitors cannot reduce the level of accumulated AGEs in the body. AGE breakers are one of the options for reducing the level of accumulated AGEs. Alagebrium chloride (ALT-711) is a well-known AGE breaker, and it breaks down the existing AGE-protein cross-links and ameliorates the pathological changes in diabetic complications [101]. AGE-protein cross-links are associated with the pathogenesis of atherosclerosis and cardiovascular diseases [102]. A recent population-based study showed that the accumulation of AGEs was independently associated with heart failure, and this association was more prominent in patients with diabetes [103]. However, an earlier clinical trial reported that alagebrium did not have a beneficial effect on the heart [104].

Another AGE cross-link breaker, TRC4186, had good tolerance and was safe in the phase I clinical trial [105], but no further studies were found.

The exact mechanism of AGE breakers has not been completely determined, and further studies are needed to understand the pharmacological mechanism of AGE cross-link breakers. There are few studies concerning the effects of AGE breakers on diabetic retinopathy. However, a recent study showed that a major dietary flavonoid, epicatechin, broke down the existing glycated albumin in vitro and reduced the accumulated AGE in the retinas in vivo [106]. Thus, epicatechin may be a potential AGE breaker and should be considered for treating diabetic retinopathy.

Curcumin (diferuloylmethane) is a representative flavonoid, and tetrahydrocurcumin, one of the major metabolites of curcumin, is a potential AGE breaker [107]. Tetrahydrocurcumin has a higher antioxidant activity than curcumin [108] and may be a stronger AGE breaker than curcumin [107]. Curcumin ameliorated both vascular and neuronal abnormalities in diabetic retinas in vivo by reducing the levels of VEGF and tumor necrosis factor-α, protecting the inner retinal barrier, reducing oxidative stress, and capillary basement membrane thickening [109,110,111]. Filippelli et al. reported that in patients with proliferative diabetic retinopathy, curcumin slightly reduced the pro-inflammatory cytokines and soluble mediators in the vitreous [112]. One of the weak points of curcumin and tetrahydrocurcumin is their low aqueous solubility which results in their poor bioavailability in the eye. However, Maharjan et al. recently developed a drug delivery system, the encapsulation of curcumin or tetrahydrocurcumin into the hydroxypropyl-cyclodextrins, which led to an increase in its solubility and enhanced the retinal epithelial permeability [113]. This system increased the antioxidant activity of the ocular epithelial cells and may become an ocular medication in the future [113].

### 5.3. RAGE Blockers and Inhibitors

AGE/RAGE signaling activates several pathways, including NADPH, Ras-ERK, Ras-MAPK, Ras-JNK, and endothelial nitric oxide synthase (eNOS). Activation of these pathways results in ROS formation, inflammatory reaction, apoptosis, and NO synthesis [9,114]. Most signals can directly or indirectly upregulate NF-κB [114] (Figure 4). Although the RAGE-independent axis cannot be blocked, RAGE blockers and inhibitors can be, in part, anti-AGE therapies for diabetic retinopathy (Figure 4).

For example, azeliragon (TTP488 or PF-4494700) is known to be a RAGE inhibitor by binding with a RAGE binding site and ameliorates Alzheimer’s disease via the Janus tyrosine kinase (JAK) and signal transducer and activator of transcription (STAT) pathway [115]. The JAK-STAT pathway is activated under the AGE/RAGE signaling and related to vascular remodeling and cellular proliferation leading to renal hypertrophy. Because there are pathophysiological pharmacotherapeutics links between Alzheimer’s disease and diabetes [116], azeliragon is one of the therapeutic options for diabetic complications. In fact, azeliragon ameliorates diabetic neuropathy in diabetic animal models [117].

### 5.4. Existing Drugs and Natural Compounds for Various Diseases

There are several existing drugs, such as glitazones, angiotensin-converting enzyme inhibitors (ACEIs), statins, or metformin, that have some effects on the AGE inhibitors. These drugs have some effects on AGE inhibitors, and thus, their anti-AGE effects may be limited, but the merit of these drugs is that they are already in use as clinical medications and their safety and tolerance have been determined.

One of the oral anti-diabetic drugs, pioglitazone, is an insulin sensitizer and is used in Japan as an additional anti-diabetic drug for patients with diabetic retinopathy. A clinical randomized controlled trial showed that pioglitazone increased the level of soluble RAGE and endogenous secretory RAGE, which are AGE/RAGE antagonists. It also decreased the expression of RAGE [118]. In addition, these anti-AGE effects were independent of the insulin resistance level [118]. Alpha-glucosidase inhibitors, another anti-diabetic drug, decreased the level of AGEs in the serum [119]. However, recent studies showed that several natural compounds inhibit α-glucosidase and AGE formation, such as *Vernonia amygdalina* extracts [120], pregnane glycosides, and the flavonoids isolated from *Carelluma hexagone* [121]; *Nymphoides indica* rhizome extracts [122], flavonoids, and proanthocyanidins-rich fractions extracted from *Eugenia dysenterica* fruits and leaves [123]; isoflavonoids isolated from *Masclura tricuspidate* [124]; isolated active constituents from *Osmanthus fragrans* flowers and structural analogs [125]; phenolics isolated from *Eugenia jambolana* [126]; or phenolics isolated from *Chrozophora oblongifolia* aerial parts [127]. Alpha-glucosidase and AGE formation is a much-discussed topic, and thus, some natural compounds that inhibit α-glucosidase and AGE formation may become commercially available as medications or supplementations in the near future.

Angiotensin II type 1 receptor blocker (ARB) and ACEI are used as major anti-hypertension drugs worldwide. Both ARB and ACEI reduce AGE formation as metal chelators and antioxidants in vitro [128] and in vivo [129,130]. The effects of anti-AGE accumulation of these drugs were shown in patients with diabetes mellitus [131,132]. A recent meta-analysis reported that the renin-angiotensin system inhibitors reduced the risk of diabetic retinopathy [133]. In addition, ACEI might be more effective than ARBs for the regression of diabetic retinopathy (registered number of International Prospective Register of Systematic Reviews, CRD42013004548) [133]. Sun et al. recently demonstrated that the ACE-mediated transforming growth factor-β1/Smad signaling pathway was associated with the blood–retinal barrier breakdown in diabetic retinopathy [134].

Other existing drugs for various diseases that have anti-AGE effects are calcium channel blockers, amlodipine and azelinidipine [135], an anti-hyperlipidemia drug, statin [136,137], and anti-diabetic drug metformin [138]. Recently, metformin was considered to be a potential therapeutic drug for neurological diseases because metformin enhances mitochondrial function by its activation of AMP-activated protein kinase (AMPK) [139]. The neuroprotective effect of metformin for patients with acute stroke and type 2 diabetes was associated with the AMPK/mammalian target of the rapamycin signaling pathway [140]. Furthermore, metformin can reduce the risk of dementia in patients with diabetes [141]. Metformin had protective effects on diabetic rat retinas by suppressing the toll-like receptor 4/NF-κB and had antioxidative stress [142]. A recent population-based cohort study reported that metformin reduced the risk of developing non-proliferative diabetic retinopathy and potentially reduced the risk of sight-threatening diabetic retinopathy [143].

Many natural compounds are known to have anti-glycation properties, but their bioactivities are relatively weak. Although there are safety profiles for natural compounds, a continuous intake as a supplement is required to prevent the development and progression of diabetic retinopathy. Several well-known natural compounds that have anti-AGE effects have been introduced in this review. For example, cinnamon bark extract inhibits AGE formation by trapping reactive carbonyls [144]. Cinnamon bark proanthocyanidins have high antioxidant and carbonyl scavenging capacity [145]. These anti-glycemic effects may be responsible for cinnamon bark’s constituents, such as catechin, epicatechin, and procyanidin B2, in improving the retinal functional abnormalities as assessed by electroretinography. They also protect the photoreceptor cells by suppressing glial activity, angiogenesis, and oxidative stress [146].

Garlic belongs to the genus *Allium,* and garlic extracts have antioxidative and free-radical scavenging effects in diabetic rats [147]. Another in vivo diabetic animal study reported that garlic reduced the total oxidative status, NO levels, and TNF-α protein levels. Therefore, garlic extracts have hypoglycemic, antioxidant, and anti-inflammatory properties [148]. The results of a recent double-blind, randomized clinical trial showed that garlic tablets improved visual acuity and decreased central macular thickness. Thus, garlic tablets can be used as a complementary treatment for diabetic retinopathy [149].

Mate tea is prepared from the dry leaves of *Ilex paraguariensis*, and it contains caffeic acid and chlorogenic acid, which are responsible for its antiglycation properties [150]. Although *Ilex paraguariensis* extracts have more anti-glycemic activities than green tea extracts [151], there are no publications regarding the effect of mate tea on diabetic retinopathy.

Green tea is made from the leaves of the *Camellia sinensis* L plant, and several polyphenol antioxidants are part of its constituents. The antioxidative effects of green tea are associated with its neuroprotective effect on diabetic retinopathy in vivo [152]. A clinic-based case-control study showed that regular Chinese green tea consumption significantly reduced the risk of developing diabetic retinopathy [153]. Similarly, a rural community-based, cross-sectional survey (*n* = 5281) demonstrated that long-term tea consumption reduced the risk of diabetic retinopathy compared to non-tea consumption [154].

AGEs are associated with neuronal cell death in several diseases, such as Alzheimer’s disease [155,156], brain ischemia [157], Parkinson’s disease [158], and diabetic retinopathy [159]. Similar to cultured retinal neurons [57,58], glyceraldehyde-derived AGEs are associated with neuronal cell death in Alzheimer’s disease [155]. A citrus flavonoid, hesperetin inhibited the ER stress-mediated neuronal apoptosis in AGEs-induced neuroblastoma cells [160]. The ER stress-mediated cell death was similarly observed in AGEs-exposed cultured retinal neurons [58,64]. Thus, hesperetin may be a potential agent for treating Alzheimer’s disease as well as diabetic retinopathy.

A bioflavonoid compound in citrus fruits, diosmetin, protected AGEs-induced neuroblastoma cell death by suppressing the ER stress-related cell death pathways [161]. Rosemary extracts are known to have comparable anti-glycation activity to aminoguanidine [162]. The results of a recent study indicated that rosmarinic acid inhibited DNA glycation, which could lead to hippocampal neuron death in diabetic animals, and it normalized the Akt1 and Akt3 expression [163]. Akt1 and Akt3 are inhibitors of apoptosis, and their expressions were downregulated in diabetic animals. Recently, Vieira et al. developed a rosmarinic acid-loaded poly lactic-co-glycolic acid intraocular implant [164]. The delivery device showed no toxic effects and reduced new vessel formation in rabbit eyes [164]. Thus, the intravitreal implant can prolong the release of rosmarinic acid and potentially can be used to prevent neovascularization in ophthalmic diseases, including diabetic retinopathy.

## 6. Conclusions

Increased AGE accumulation induces not only vascular abnormalities but also neuronal abnormalities, and it is involved in several retinal and optic nerve diseases, including diabetic retinopathy, age-related macular degeneration, and glaucoma. In addition, AGEs are associated with neuronal diseases such as Alzheimer’s disease, Parkinson’s disease, and strokes. Anti-AGE therapies are one of the options for preventing the dysfunction of the NVUs of diabetic retinopathy. The first-line anti-AGE therapy is reducing the intake of exogenous AGEs. Thus, low-AGE food products are practically selected for patients with diabetes. However, chronic hyperglycemia leads to endogenous AGE accumulation in the body of diabetic patients, which results in the development of diabetic complications. Therefore, anti-AGE drugs should be required to reduce endogenous accumulated AGEs. The anti-AGE drugs should have the effects of antioxidants, metal chelators, and carbonyl scavengers. In addition, cross-link breakers and RAGE inhibitors are also potential drugs for anti-AGE therapies. However, there are no ideal anti-AGE drugs with significant therapeutic effects and without any side effects. Natural compounds have lower bioactivities for anti-AGE effects than chemical compounds, but they have the merits of their safety profiles. Some natural compounds for anti-AGE therapies may be useful for the treatment of patients with early diabetic retinopathy. Further basic and clinical studies should be performed to establish the effectiveness and safety of anti-AGE drugs for diabetic retinopathy.

## Figures and Tables

**Figure 1 ijms-24-02927-f001:**
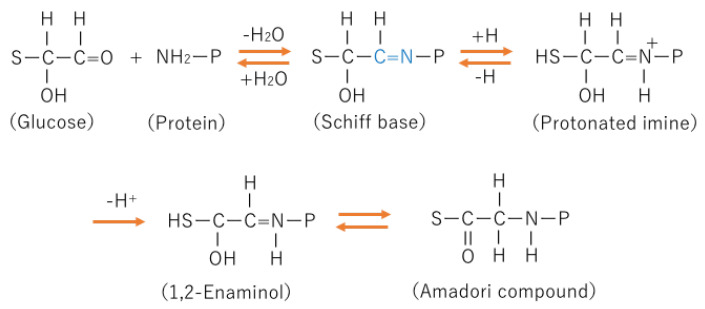
Initial step of the Maillard reaction. In the first step, a protein reacts with an amino acid, which results in the production of a Schiff base. The Schiff base is converted into an Amadori product by the Amadori rearrangement through a protonated imine and 1,2-enaminol formation. S: reducing sugar, P: protein.

**Figure 2 ijms-24-02927-f002:**
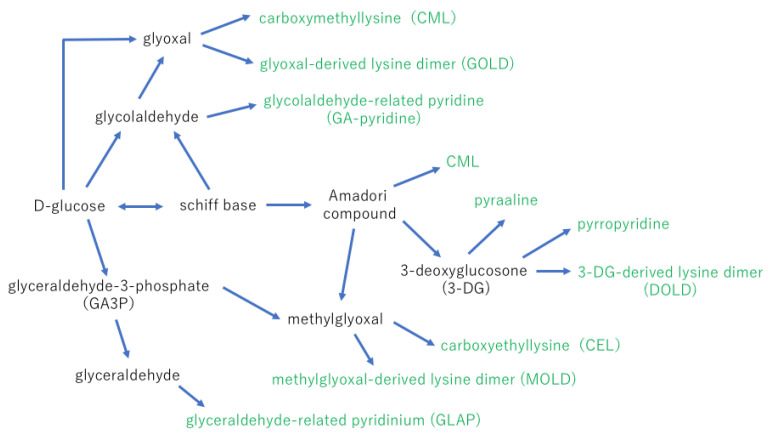
Hypothetical schemes of the final step of AGE formation. Carboxymethyllysine (CML) is synthesized from an Amadori compound or glyoxal [23]. Carboxyethyllysine (CEL) is made from methylglyoxal [24]. Glyoxal-derived lysine dimer (GOLD), methylglyoxal-derived lysine dimer (MOLD), and 3-DG-derived lysine dimer (DOLD) have cross-links between lysine residues and glyoxal, methylglyoxal or 3-DG [25]. Only a small percentage of AGEs in the body have been identified. Glyceraldehyde- and glycolaldehyde-derived AGEs may be synthesized faster than Amadori compound-derived AGEs and more toxic than other types of AGEs.

**Figure 3 ijms-24-02927-f003:**
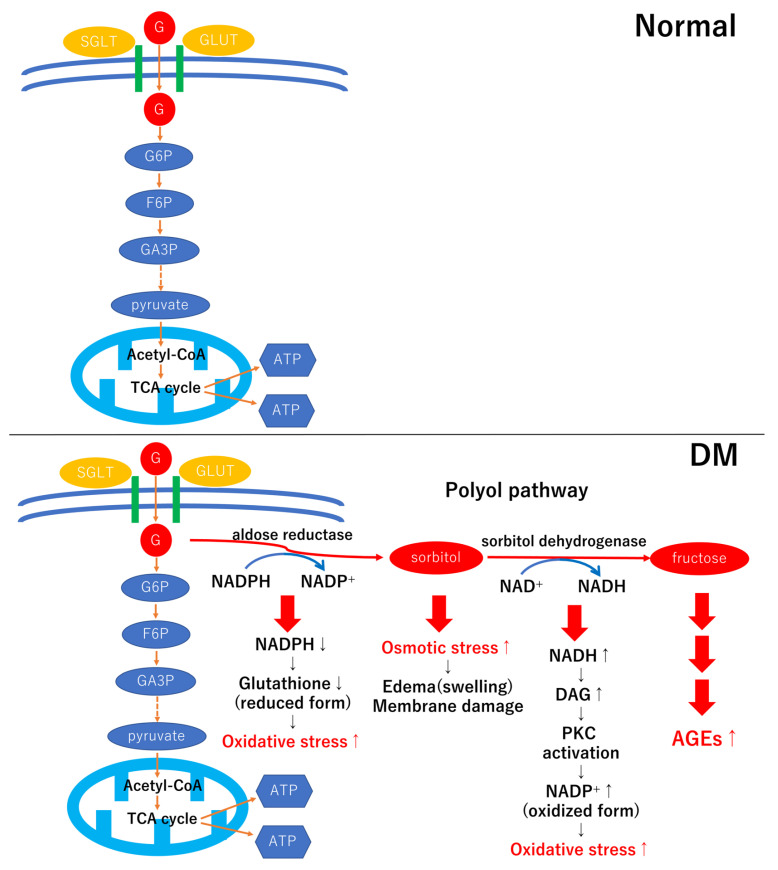
Scheme of the polyol pathway. Under normal conditions (upper figure), glucose is taken in by the retinal cells via sodium-glucose cotransporter (SGLT) or glucose transporter (GLUT). Glucose is converted to pyruvate by the metabolic glycolysis pathway. This process releases free energy to synthesize the high-energy molecule adenosine triphosphate (ATP) in the final step. In the diabetic condition, excessive uptake of glucose activates a minor cellular pathway, the polyol pathway (lower figure). The activated polyol pathway causes an increase in oxidative stress, osmotic stress, and AGE formation in diabetic retinas. G: glucose, G6P: glucose-6-phosphate, F6P: fructose-6-phosphate, GA3P: glyceraldehyde-3-P, NADP: nicotinamide adenine dinucleotide phosphate (NADP+; oxidized form, NADPH; reduced form), DAG: diacylglycerol, PKC: protein kinase C, AGEs: advanced glycation end products.

**Figure 4 ijms-24-02927-f004:**
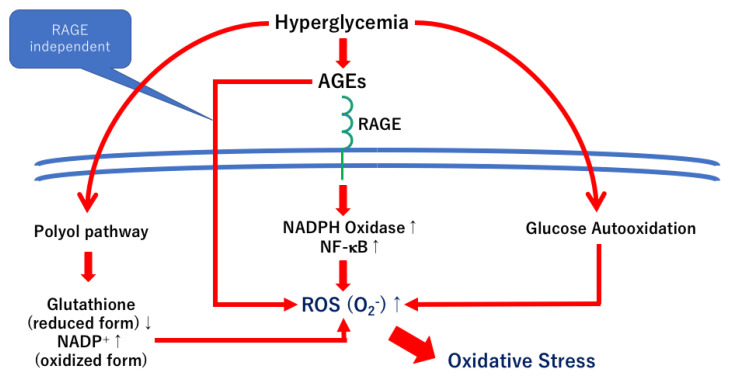
Hypothetic scheme of the pathophysiology of AGEs in diabetic retinas. Chronic hyperglycemia facilitates AGE production. Oxidative stress is induced by the AGE-RAGE axis, but Amadori compounds become the source of ROS production. Thus, the RAGE-independent pathways can be involved in increasing oxidative stress in diabetic retinas.

## Data Availability

Not applicable.

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
