# Peer review of "Advanced Glycation End-Products and Diabetic Neuropathy of the Retina"

_ijms, 2023, doi:10.3390/ijms24032927_

Round 1
Reviewer 1 Report
The review "Advanced glycation end-products and diabetic neuropathy of the retina" discusses the role of ACE compounds and their role in diabetic retinopathy. The review is well written and presents a comprehensive account of the topic.
I have one suggestion, according to the authors several pathways that was mentioned in the review as pathways involved in AGE generation like RAS, ERK, MAPK, PKC and JNK are also very important in several different cancers and inhibitors of these pathways have been extensively studied in context of cancer with multiple preclinical and clinical studies. It would be nice to have a chapter exploring the overlap of these drugs as AGE therapies and the information available (if any) of usage of those pathway blockers as potential AGE therapeutics.
Reviewer 2 Report
Oshitari realized a very interesting review describing the “Advanced glycation end-products and diabetic neuropathy of the retina”. I consider the manuscript very interesting but, at the same time, I suggest several revisions needed to improve the reliability and the completeness of the paper:
· The “Introduction” sections should be more updated and improved. I suggest adding data related to the involvement of A2E and their possibly related components, especially with vascular ones. The recent PMID: 32877751, PMID: 30523548, PMID: 36490268 and PMID: 36290689 could represent a substrate able to enforce the role of considered cellular mechanisms.
· Finally, manuscript requires important English revisions and typos correction.
Round 2
Reviewer 2 Report
The manuscript can be accepted in present form.